

# Evaluation of procedures for typing of group B *Streptococcus*: a retrospective study

Hans-Christian Slotved and Steen Hoffmann

Neisseria and Streptococcus Reference Laboratory, Department of Microbiology and Infection Control, Statens Serum Institut, Copenhagen, Denmark

## ABSTRACT

**Background.** This study evaluates two procedures for typing of *Streptococcus agalactiae* (group B streptococci; GBS) isolates, using retrospective typing data from the period 2010 to 2014 with a commercial latex agglutination test (latex test) and the Lancefield precipitation test (LP test). Furthermore, the genotype distribution of phenotypically non-typable (NT) GBS isolates is presented. We also raise the awareness, that the difference in typing results obtained by phenotypical methods and genotype based methods may have implications on vaccine surveillance in case a GBS vaccine is introduced.

**Methods.** A total of 616 clinical GBS isolates from 2010 to 2014 were tested with both a latex test and the LP test. Among these, 66 isolates were genotyped by PCR, including 41 isolates that were phenotypically NT.

**Results.** The latex test provided a serotype for 83.8% of the isolates (95% CI [80.7–86.6]) compared to 87.5% (95% CI [84.6–90.0]) obtained by the LP method. The two assays provided identical capsular identification for all sero-typeable isolates (excluding NT isolates). The PCR assay provided a genotype designation to the 41 isolates defined as phenotypically NT isolates.

**Discussion.** We found that the latex test showed a slightly lower identification percentage than the LP test. Our recommendation is to use the latex agglutination as the routine primary assay for GBS surveillance, and then use the more labour intensive precipitation test on the NT isolates to increase the serotyping rate. A genotype could be assigned to all the phenotypically NT isolates, however, as a consequence genotyping will overestimate the coverage from possible future capsular polysaccharide based GBS vaccines.

## INTRODUCTION

*Streptococcus agalactiae* (group B streptococcus, GBS) is a well-known pathogen primarily causing infections in newborns and the elderly (*Brigtsen et al., 2015*; *Ballard et al., 2016*). The disease in neonates is generally described as occurring in two different varieties (*Bulkowstein et al., 2016*). Early-onset disease (EOD) occurs in the neonate during the first six days of life, while late-onset disease (LOD) occurs later than the seventh day of life and can develop up to three months of age (*Le Doare & Heath, 2013*; *Vinnemeier et al., 2015*). Possible

Corresponding author
Hans-Christian Slotved, hcs@ssi.dk

clinical manifestations of GBS infection in neonates are sepis, meningitis and pneumonia (*Schrag & Verani, 2013*). Among adults, GBS may also be associated with invasive infections, particularly in elderly persons with underlying medical conditions (*Le Doare & Heath, 2013*). Since introduction of screening programmes for pregnant women (*Ballard et al., 2016*) in some parts of the developed world, particularly early onset GBS infections in neonates has been reduced, and for many years the GBS disease incidence has been low (*Heath, 2016*). In contrast, early onset GBS infection is still a major problem in the developing world and presumably an underestimated problem (*Heath, 2016*). Also in recent years, the developed world has seen an increasing interest in incidence of invasive GBS infections, in particular among the elderly (*Sheppard et al., 2016*). Surveillance and identification of GBS in humans are therefore increasingly essential (*Ballard et al., 2016*; *Sheppard et al., 2016*).

The GBS are currently divided into ten serotypes based on type specific capsular antigens and are designated as Ia, Ib, II, III, IV, V, VI, VII, VIII, and IX (*Slotved et al., 2007*; *Le Doare & Heath, 2013*). For decades, the precipitation test also known as the Lancefield precipitation test (LP test) has been considered the standard method for GBS serotype determination (*Slotved, Sauer & Konradsen, 2002*). However, the method is time-consuming and therefore not suited for typing large numbers of isolates (*Slotved et al., 2003*). At present, GBS isolates are in general serotyped by the phenotypical method latex agglutination test (latex test) (*Afshar et al., 2011*), for which several kits are commercially available. Increasingly simpler and affordable molecular techniques for genotyping of GBS isolates, predominantly based on PCR assays, have been introduced and are now commonly used (*Brigtsen et al., 2015*; *Sheppard et al., 2016*).

In recent years, all GBS isolates received at the Statens Serum Institut (SSI) have been typed using both a LP test and a latex test (*Slotved, Sauer & Konradsen, 2002*; *Lambertsen et al., 2010*). Furthermore, some of the GBS NT isolates have been tested for genotype using the PCR assay described by *Imperi et al. (2010)* and *Poyart et al. (2007)*.

By using all our GBS typing data from the period 2010 to 2014, we evaluated the phenotypical typing procedure for GBS isolates, based on the comparison of the commercial latex test and the LP test. We will furthermore show the genotype distribution of phenotypically NT GBS isolates.

## METHODS

This is a retrospective study based on typing data obtained in the period from 2010 to 2014 at the national Neisseria and *Streptococcus* Reference Laboratory (NSR), SSI.

The Danish hospitals are serviced by regional departments of clinical microbiology, all of which are public. On a voluntary basis, they submit isolates of beta-hemolytic streptococci to the NSR for national surveillance (*Lambertsen et al., 2010*).

### Isolates

The study was based on 616 isolates received at the NSR laboratory (SSI) in the period of 2010–2014. The majority of the isolates were from bloodstream infections and each isolate represented one patient case.

The LP test is considered the reference method at the NSR laboratory (SSI) (*Slotved, Sauer & Konradsen, 2002*; *Lambertsen et al., 2010*).

## Identification of GBS isolates

The GBS isolates were identified as described by *Lambertsen et al. (2010)*. Briefly, the submitted strains were examined for their characteristic beta-hemolytic colonies on 5% horse blood agar plates (SSI Diagnostica, Hillerød, Denmark) followed by serogrouping with group B latex (Oxoid A/S, Greve, Denmark) as recommended by the manufacturer. Isolates were stored at −80 °C in nutrient beef broth containing 10% glycerol (SSI Diagnostica, Hillerød, Denmark).

## Serotyping of the GBS isolates

All isolates were tested both with the LP test and latex test (SSI Diagnostica, Hillerød, Denmark) (*Lambertsen et al., 2010*).

## Lancefield precipitation test (LP test)

The precipitation test was performed as described by *Slotved, Sauer & Konradsen (2002)*. Briefly, a resuspended centrifuged overnight broth culture was boiled and treated with 0.2N hydrochloric acid (0.2N HCl) to extract the capsular antigen. A LP test was performed, by mixing the extract with serotype specific GBS antisera (Ia–IX) (SSI Diagnostica, Hillerød, Denmark). If no reaction occurred then an extract with 0.1N HCl was made and tested. Non-serotypeable isolates were designated NT.

See *SSI Diagnostica (2017)* for a video description of the LP test.

## Latex agglutination test (latex test)

The latex test was performed with the *Streptococcus* latex test ImmuLex™ (SSI Diagnostica, Hillerød, Denmark). Briefly, isolates were cultured for 24 h in Todd-Hewitt broth. Ten microlitres from this culture was mixed with 10 microlitres specific antisera corresponding to one of each of serotypes Ia, Ib, and II–IX specific to capsular polysaccharide antigen latex test suspension, and agglutination was read after 5–10 s (*Slotved et al., 2003*).

## PCR test

The multiplex PCR assay and primers (TAG Copenhagen) used in this study were described by *Imperi et al. (2010)* and *Poyart et al. (2007)*. Briefly, 0.5 ml Chelex solutions were prepared of each isolate. The multiplex PCR for the genes was performed using a 20 μl PCR mix of 10 μl HotstarTaq Mastermix (Qiagen, Hamburg, Germany). The following PCR program used was: 15 min at 95 °C, 35 cycles of 15 s at 95 °C, 50 s at 55 °C, 60 s at 72 °C, finalized with 10 min at 72 °C. The presence and quality of expected PCR fragments were tested by gel-electrophoresis on 2% E-gels (Invitrogen).

In the years 2010 and 2011 we chose to evaluate the PCR on a majority of the phenotypically NT isolates.

## RESULTS

### Serotyping by the LP test (0.1N HCl and 0.2N HCl) (Table 1)

Of the 616 GBS isolates tested with 0.2N HCl, it was possible to serotype 530 isolates (86.0%, (95% CI [83.0–88.6])). The 86 isolates that did not show a serotype reaction were tested using 0.1N HCl, thereby identifying further nine isolates, resulting in a total identification of 539 isolates (87.5%, (95% CI [84.6–90.0])). The remaining 77 isolates (12.5%, (95% CI [10.0–15.4])) were classified as NT isolates. It was predominantly serotype III (3/185 isolates) and VIII isolates (3/15) which were identified with 0.1N HCl. One isolate of each of serotype Ia (1/113 isolates), serotype V (1/77 isolates), and serotype XI (1/5 isolates) were identified using 0.1N HCl.

### A comparison of latex test with the LP test (Table 1)

With the latex test it was possible to serotype 516 isolates (83.8%, (95% CI [80.7–86.6])), while 100 isolates (16.2%, (95% CI [13.4–19.4])) could not be identified, due to either multiple reactions (cross-reactions) or no reactions. The latex test provided results identical to those obtained by the LP test, except for 11 isolates that were NT by the LP test. Six of these 11 isolates were serotype V. Among 100 isolates that were NT with the latex test, 34 were serotyped with the LP test (Table 1).

### Molecular typing by PCR

A total of 66 of the 616 isolates were tested for their genotype. Of these isolates 25 could be assigned a serotype, while 41 isolates were considered serotype NT (Tables 2 and 3). The 41 NT isolates included nearly all NT isolates for 2010 (24 of total 30) and 2011 (15 of total 16) (Table 2). The genotyping of the 41 isolates designated as NT by the LP test, showed a high predominance of serotype V (15 isolates) followed by serotype III (9 isolates) (Table 2). Nearly all phenotypically NT isolates could be genotyped (Table 2).

When comparing the genotype and serotype by LP test of the 25 identified isolates, a difference was noted for two isolates. Both isolates were genotype II, while they were serotype III with 0.2N HCl (Table 3). Two isolates identified as genotype V were identified as serotype VI and VII with the LP test, while they were NT with the LP test (Table 4).

The combined vaccine relevant serotypes (serotype Ia, Ib and III) represented 49.7% (95% CI [42.0–57.4]) of phenotyped isolates in 2010 and 56.6% (95% CI [47.8–65.1]) in 2011. If those phenotype NT isolates, where a genotype of one of the three vaccine serotypes was established, were added to the serotype distribution, then the vaccine relevant serotypes (serotype Ia, Ib and III) represented 56.6% (95% CI [48.9–64.1]) in 2010 and 61.0% (95% CI [52.3–69.2]) in 2011.

## DISCUSSION

In general, laboratories use latex tests, and to a minor extent LP test for GBS serotyping (*Sheppard et al., 2016*; *Afshar et al., 2011*). In this study, we found the latex test able to serotype 83.8% (95% CI [80.7–86.6]), while the LP test was able to serotype 87.5% (95% CI [84.6–90.0]) of the isolates (Table 1). Both assays provided serotype identification to some isolates that were non-typeable with the other method, although to a varying degree

**Table 1  Comparison of Lancefield precipitation test (LP test) (0.2N HCl + 0.1N HCl) and latex test.**

| Serotype assigned by latex test[b] | Serotype assigned by LP test[a] | | | | | | | | | | | |
|---|---|---|---|---|---|---|---|---|---|---|---|---|
| | Ia | Ib | II | III | IV | V | VI | VII | VIII | IX | NT | Total number |
| Ia | 105 | | | | | | | | | | 1 | 106 |
| Ib | | 49 | | | | | | | | | | 49 |
| II | | | 57 | | | | | | | | 1 | 58 |
| III | | | | 172 | | | | | | | | 172 |
| IV | | | | | 24 | | | | | | | 24 |
| V | | | | | | 73 | | | | | 6 | 79 |
| VI | | | | | | | 4 | | | | 1 | 5 |
| VII | | | | | | | | 1 | | | 1 | 2 |
| VIII | | | | | | | | | 15 | | | 15 |
| IX | | | | | | | | | | 5 | 1 | 6 |
| NT | 8 | 4 | 3 | 13 | 2 | 4 | | | | | 66 | 100 |
| Total number (0.2N HCl + 0.1N HCl) | 113 (112 + 1) | 53 (53 + 0) | 60 (60 + 0) | 185 (182 + 3) | 26 (26 + 0) | 77 (76 + 1) | 4 (4 + 0) | 1 (1 + 0) | 15 (12 + 3) | 5 (4 + 1) | 77 (77 + 0) | 616 (607 + 9) |

**Notes.**

[a] If Ia to IX 0.2N HCl was negative, then automatically Ia to IX 0.1N HCl was tested.

[b] If cross-reactions were observed with the Latex test, the isolate was considered to be non-typeable.

**Table 2 Results of genotyping of 39 phenotypically non-typeable GBS isolates in 2010 and 2011.**

| Serotype assigned by Lancefield precipitation test[a] | Genotype assigned by PCR | | | | | | | | | | | | |
|---|---|---|---|---|---|---|---|---|---|---|---|---|---|
| | Ia | Ib | II | III | IV | V | VI | VII | VIII | IX | NT | Total number of genotyped NT isolates | No PCR |
| 2010 (143/173 isolates tested) | 4 | 1 | 1 | 7 | | 10 | | | 1 | | | 24 of total 30 | 5 |
| 2011 (120/136 isolates tested) | 1 | 3 | 3 | 2 | | 4 | | | 1 | 1 | | 15 of total 16 | 1 |
| Total number of isolates of each genotype | 5 | 4 | 4 | 9 | 0 | 14 | 0 | 0 | 2 | 1 | 0 | | 6 |

Notes.
[a]Phenotypical identification excluding NT/total number of isolates. If Ia to IX 0.2N HCl was negative, then automatically Ia to IX 0.1N HCl was tested.

**Table 3 Comparison of Lancefield precipitation test (LP test) results with the PCR for 66 isolates.**

| Serotype assigned by LP test[a] | Genotype assigned by PCR | | | | | | | | | | | |
|---|---|---|---|---|---|---|---|---|---|---|---|---|
| | Ia | Ib | II | III | IV | V | VI | VII | VIII | IX | NT | Total number |
| Ia | 4 | | | | | | | | | | | 4 |
| Ib | | 2 | | | | | | | | | | 2 |
| II | | | 2 | | | | | | | | | 2 |
| III | | | 2 | 7 | | | | | | | | 9 |
| IV | | | | | 2 | | | | | | | 2 |
| V | | | | | | 5 | | | | | | 5 |
| VI | | | | | | | 1 | | | | | 1 |
| VII | | | | | | | | | | | | 0 |
| VIII | | | | | | | | | | | | 0 |
| IX | | | | | | | | | | | | 0 |
| NT | 5 | 5 | 4 | 9 | | 15 | | | 2 | 1 | | 41 |
| Total number | 9 | 7 | 8 | 16 | 2 | 20 | 1 | 0 | 2 | 1 | 0 | 66 |

Notes.
[a]If no reaction with Ia to IX was found with 0.2N HCl, then Ia to IX were tested with 0.1N HCl.
Red number represent isolates that were identified with contradicting typing.

(Table 1). We did not find any conflicting test results using the two assays except for the NT isolates. A serotyping identification rate between 80% and 90% is common (*Brigtsen et al., 2015*), and even lower serotyping rates have been observed (*Slotved et al., 2003*; *Yao et al., 2013*). Because the latex test is much easier to perform than the LP test, there is no question on which phenotypical method to use for routine serotyping (*Brigtsen et al., 2015*; *Sheppard et al., 2016*). At the NSR laboratory (SSI) we have chosen the following procedure for phenotypical serotyping (Fig. 1): we start with the latex test, if this method provide serotype identification, then the identification is accepted, and no further serotype testing is performed. If the latex test shows either cross-reactions or non-typeability, then we proceed to test the isolate using the LP test, by first testing for 0.2N HCl and then if necessary proceed to 0.1N HCl. If the LP test provides a specific serotype, then this is

**Table 4  Comparison of Latex test result with the PCR.**

| Serotype assigned by latex test[a] | Genotype assigned by PCR | | | | | | | | | | | |
| --- | --- | --- | --- | --- | --- | --- | --- | --- | --- | --- | --- | --- |
| | Ia | Ib | II | III | IV | V | VI | VII | VIII | IX | NT | Total number |
| Ia | 4 | | | | | | | | | | | 4 |
| Ib | | | | | | | | | | | | 0 |
| II | | | | | | | | | | | | 0 |
| III | | | | 3 | | | | | | | | 3 |
| IV | | | | | 1 | | | | | | | 1 |
| V | | | | | | 6 | | | | | | 6 |
| VI | | | | | | 1 | 1 | | | | | 2 |
| VII | | | | | | 1 | | | | | | 1 |
| VIII | | | | | | | | | | | | 0 |
| IX | | | | | | | | | | | | 0 |
| NT | 5 | 7 | 8 | 13 | 1 | 12 | | | 2 | 1 | | 49 |
| Total number | 9 | 7 | 8 | 16 | 2 | 20 | 1 | 0 | 2 | 1 | 0 | 66 |

**Notes.**
[a]If cross-reactions were observed with the Latex test, the isolate was considered to be non-typeable.
Red numbers represent two isolates that were non-typeable with 0.2N HCl and 0.1N HCl.

accepted, or else the isolate is defined as non-typeable. Using this procedure provides a serotyping percentage of 89.3% (95% CI [86.6–91.6]) (Fig. 1).

In recent years several studies (*Brigtsen et al., 2015*; *Sheppard et al., 2016*) have presented molecular based methods for typing of GBS isolates, and a standard GBS PCR method has been described (*Imperi et al., 2010*). The most recent molecular GBS typing method described is the Whole-Genome Sequencing, which has the advantage, that besides providing information on the capsular genes, it also can provide information on multilocus sequence type, analyses of relatedness to other sequenced isolates, and detailed phylogenetic analyses (*Sheppard et al., 2016*).

Our PCR assay provided a genotype for all the 41 phenotypically NT isolates (Table 2). According to other studies, nearly 100% of GBS isolates can be genotyped by the use of molecular based methods (*Brigtsen et al., 2015*; *Yao et al., 2013*; *Sheppard et al., 2016*). This typing rate is much higher than the approximately 90% rate obtained by phenotypical assays (*Brigtsen et al., 2015*) (Table 1).

The two GBS vaccines currently under Phase 2 trials are based on capsular polysaccharide conjugated vaccines, while another vaccine based on GBS surface proteins is under Phase 1 trial (*Heath, 2016*). The capsular polysaccharide based vaccines cover either serotype III or serotype Ia, Ib and IIII (*Heath, 2016*). Evaluating the predicted vaccine coverage for Danish invasive GBS isolates from 2010 and 2011 in this study, we found that genotyping suggested an apparent increase in predicted vaccine coverage of 6.9% in 2010 and 4.4% in 2011. However, as these isolates were only typeable by molecular methods, the type identification represents lack of phenotypical expression and therefore possible lack of vaccine coverage.

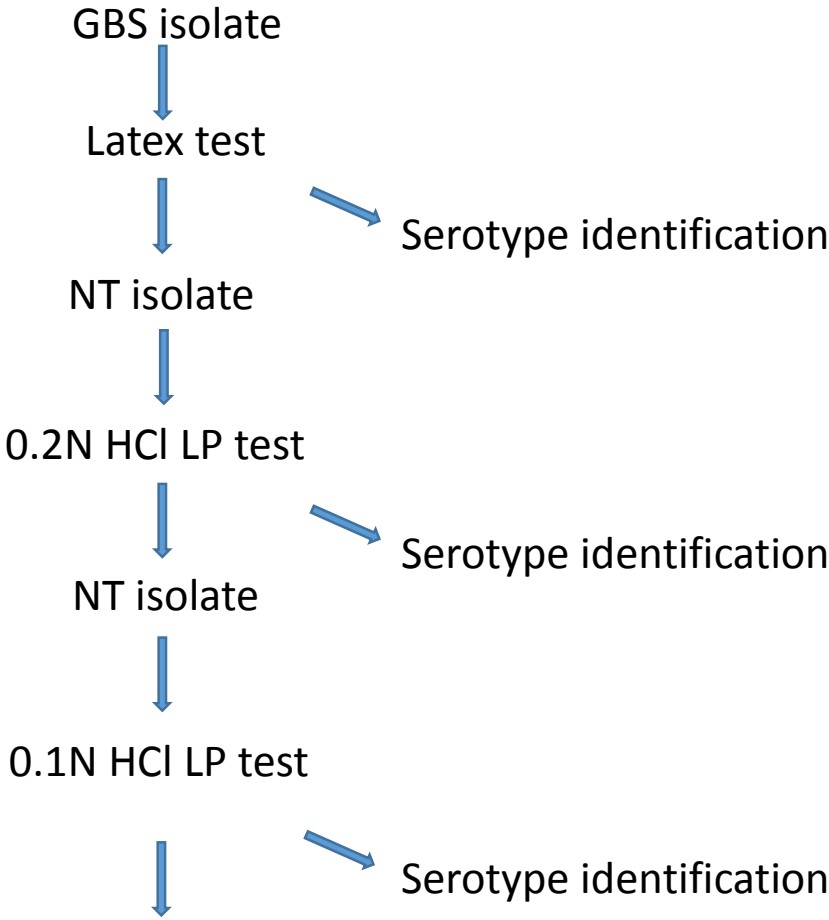

GBS isolate

Latex test

Serotype identification

NT isolate

0.2N HCl LP test

Serotype identification

NT isolate

0.1N HCl LP test

Serotype identification

## Isolate defined as phenotypical NT isolate

**Figure 1** **A description of the GBS phenotypical serotype identification procedure at Statens Serum Institut.**

In conclusion, in this study we found that the latex test and the LP test showed similar identification percentages. Because of the greater workload with LP test, we recommend this method to be used only for latex test NT isolates (Fig. 1).

Molecular typing methods are advantageous for the surveillance of GBS infections in terms of evaluating transmission chains and possible description of, e.g., early onset neonatal infections (*Bergseng et al., 2009*). In contrast, phenotypical methods must be applied when evaluating possible vaccine coverage or vaccine failures as well as planning of future capsular polysaccharide vaccines. Therefore, appropriate typing methods must be chosen according to the purpose of surveillance. Based on our findings, we suggest that general surveillance can be performed either by using the phenotypical procedure shown in Fig. 1 or by molecular techniques such as those described by *Sheppard et al. (2016)*, depending on the laboratory capacity and cost. Coverage studies to provide data for possible future polysaccharide based GBS vaccines will require a phenotypical procedure. In case a

polysaccharide GBS vaccine will be implemented in the future, phenotypical procedures will be necessary when evaluating vaccine failure in patients with infections caused by vaccine serotypes. Molecular techniques including Multilocus sequence typing (MLST) are necessary if information on clonal relation is needed, e.g., for identification of transmission chains in case of outbreaks or clustered infections.

## ACKNOWLEDGEMENTS

Kirsten Burmeister and Monja Hammer are acknowledged for their skilful laboratory work and input to this study. We acknowledge the Danish departments of clinical microbiology for submitting *Streptococcus* group B isolates for national surveillance throughout the study period.

### Funding
The authors received no funding for this work.

### Competing Interests
The authors declare there are no competing interests.

### Author Contributions
- Hans-Christian Slotved conceived and designed the experiments, performed the experiments, analyzed the data, contributed reagents/materials/analysis tools, wrote the paper, prepared figures and/or tables, reviewed drafts of the paper.
- Steen Hoffmann contributed reagents/materials/analysis tools, wrote the paper, prepared figures and/or tables, reviewed drafts of the paper.

### Data Availability
   The raw data has been supplied as a Data S1.

### Supplemental Information
Supplemental information for this article can be found online at http://dx.doi.org/10.7717/peerj.3105#supplemental-information.

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
