# Peer review of "Evaluation of procedures for typing of group B Streptococcus: a retrospective study"

_PeerJ, doi:10.7717/peerj.3105_

## Round 0.1 · original submission · Minor Revisions

Both reviewers agree that this is a good manuscript that only needs minor revisions.

Reviewer 1 ·

Basic reporting

This reviewer found some minor errors, that should be revised for a better understanding of the text. Detected text that needs correction, are marked in yellow in the attached PDF of the revised manuscript.

Experimental design

No comments.

Validity of the findings

Providing scientific evidence to compare methodologies commonly used in clinic, is continously needed to improve health services and to support epidemiological data. In this regard, we find this type of work necessary in scientific publications.

However, since this study does not try to fill a gap in the knowledge of testing methods of genotyping GBS, but to evaluate available techniques that could enlight decision making, this reviewer considers important to state or declare how this study allows health-care scientific community a science based criteria for diagnosis or vaccine surveillance.

More specifically, this reviewr thinks it would be better for readers to find a proposal, as result of your findings, of how and when to choose a GBS typing method, well stated at the end of your conclusions. For example:
- When surveillance purpose is predicting... it is recommended to genotype ... to assure that,
- When vaccine surveillance is containing a spread, we recommend the GBS serotyping since it will provide more accurate data that will help decision making...
- When a patient in the hospital has shown a GBS infection... it is recommended to apply "X" typing method ... because...

Maybe Figure 1, can be modified to add these recommendations to a better interpretation of conclusions, through the flowchart.

Annotated reviews are not available for download in order to protect the identity of reviewers who chose to remain anonymous.

Reviewer 2 ·

Basic reporting

Name of Manuscript “Evaluation of procedures for typing of group B Streptococcus”

Hans-Christian Slotved and Steen Hoffmann evaluated 616 clinical GBS isolates tested with two procedures: latex agglutination test and the Lancefield precipitation test for typing of Streptococcus agalactiae isolates, using retrospective data from the period 2010 to 2014. Both technics are used for Serotyping GBS isolates.These methods are easier , faster and can identified GBS and not require specialized equipment.
Is important to write more introduction and background , for example, about the disease early onset and the diseases late onset ( 2002, Palacios-Saucedo et al)

English is clear and professional

The references are few, you can use more references.

Professional article structure, tables, ok

Experimental design

The submission is clear

Methods ok

Validity of the findings

Is important to write more introduction and background , for example, about the disease early onset and the diseases late onset ( 2002, Palacios-Saucedo et al)


The references are few, you can use more references.

Annotated reviews are not available for download in order to protect the identity of reviewers who chose to remain anonymous.

---

## Round 0.2 · accepted · Accept

All the suggestions were clearly done, the paper will be useful for medical purposes.